# Deep Probabilistic Programming

**Dustin Tran**
Columbia University

**Matthew D. Hoffman**
Adobe Research

**Rif A. Saurous**
Google Research

**Eugene Brevdo**
Google Brain

**Kevin Murphy**
Google Research

**David M. Blei**
Columbia University

## Abstract

We propose Edward, a Turing-complete probabilistic programming language. Edward defines two compositional representations—random variables and inference. By treating inference as a first class citizen, on a par with modeling, we show that probabilistic programming can be as flexible and computationally efficient as traditional deep learning. For flexibility, Edward makes it easy to fit the same model using a variety of composable inference methods, ranging from point estimation to variational inference to MCMC. In addition, Edward can reuse the modeling representation as part of inference, facilitating the design of rich variational models and generative adversarial networks. For efficiency, Edward is integrated into TensorFlow, providing significant speedups over existing probabilistic systems. For example, we show on a benchmark logistic regression task that Edward is at least 35x faster than Stan and 6x faster than PyMC3. Further, Edward incurs no runtime overhead: it is as fast as handwritten TensorFlow.

## 1 Introduction

The nature of deep neural networks is compositional. Users can connect layers in creative ways, without having to worry about how to perform testing (forward propagation) or inference (gradient-based optimization, with back propagation and automatic differentiation).

In this paper, we design compositional representations for probabilistic programming. Probabilistic programming lets users specify generative probabilistic models as programs and then "compile" those models down into inference procedures. Probabilistic models are also compositional in nature, and much work has enabled rich probabilistic programs via compositions of random variables (Goodman et al., 2012; Ghahramani, 2015; Lake et al., 2016).

Less work, however, has considered an analogous compositionality for inference. Rather, many existing probabilistic programming languages treat the inference engine as a black box, abstracted away from the model. These cannot capture probabilistic inferences that reuse the model's representation—a key idea in recent advances in variational inference (Kingma & Welling, 2014; Rezende & Mohamed, 2015; Tran et al., 2016b), generative adversarial networks (Goodfellow et al., 2014), and also in more classic inferences (Dayan et al., 1995; Gutmann & Hyvärinen, 2010).

We propose Edward[1], a Turing-complete probabilistic programming language which builds on two compositional representations—one for random variables and one for inference. By treating inference as a first class citizen, on a par with modeling, we show that probabilistic programming can be as flexible and computationally efficient as traditional deep learning. For flexibility, we show how Edward makes it easy to fit the same model using a variety of composable inference methods, ranging from point estimation to variational inference to MCMC. For efficiency, we show how to integrate Edward into existing computational graph frameworks such as TensorFlow (Abadi et al., 2016). Frameworks like TensorFlow provide computational benefits like distributed training, parallelism, vectorization, and GPU support "for free." For example, we show on a benchmark task that Edward's Hamiltonian Monte Carlo is many times faster than existing software. Further, Edward incurs no runtime overhead: it is as fast as handwritten TensorFlow.

---

[1]See Tran et al. (2016a) for details of the API. A companion webpage for this paper is available at `http://edwardlib.org/iclr2017`. It contains more complete examples with runnable code.

## 2 RELATED WORK

Probabilistic programming languages (PPLs) typically trade off the expressiveness of the language with the computational efficiency of inference. On one side, there are languages which emphasize expressiveness (Pfeffer, 2001; Milch et al., 2005; Pfeffer, 2009; Goodman et al., 2012), representing a rich class beyond graphical models. Each employs a generic inference engine, but scales poorly with respect to model and data size. On the other side, there are languages which emphasize efficiency (Spiegelhalter et al., 1995; Murphy, 2001; Plummer, 2003; Salvatier et al., 2015; Carpenter et al., 2016). The PPL is restricted to a specific class of models, and inference algorithms are optimized to be efficient for this class. For example, Infer.NET enables fast message passing for graphical models (Minka et al., 2014), and Augur enables data parallelism with GPUs for Gibbs sampling in Bayesian networks (Tristan et al., 2014). Edward bridges this gap. It is Turing complete—it supports any computable probability distribution—and it supports efficient algorithms, such as those that leverage model structure and those that scale to massive data.

There has been some prior research on efficient algorithms in Turing-complete languages. Venture and Anglican design inference as a collection of local inference problems, defined over program fragments (Mansinghka et al., 2014; Wood et al., 2014). This produces fast program-specific inference code, which we build on. Neither system supports inference methods such as programmable posterior approximations, inference models, or data subsampling. Concurrent with our work, WebPPL features amortized inference (Ritchie et al., 2016). Unlike Edward, WebPPL does not reuse the model's representation; rather, it annotates the original program and leverages helper functions, which is a less flexible strategy. Finally, inference is designed as program transformations in Kiselyov & Shan (2009); Ścibior et al. (2015); Zinkov & Shan (2016). This enables the flexibility of composing inference inside other probabilistic programs. Edward builds on this idea to compose not only inference within modeling but also modeling within inference (e.g., variational models).

## 3 COMPOSITIONAL REPRESENTATIONS FOR PROBABILISTIC MODELS

We first develop compositional representations for probabilistic models. We desire two criteria: (a) integration with computational graphs, an efficient framework where nodes represent operations on data and edges represent data communicated between them (Culler, 1986); and (b) invariance of the representation under the graph, that is, the representation can be reused during inference.

Edward defines random variables as the key compositional representation. They are class objects with methods, for example, to compute the log density and to sample. Further, each random variable $\mathbf{x}$ is associated to a tensor (multi-dimensional array) $\mathbf{x}^*$, which represents a single sample $\mathbf{x}^* \sim p(\mathbf{x})$. This association embeds the random variable onto a computational graph on tensors.

The design's simplicity makes it easy to develop probabilistic programs in a computational graph framework. Importantly, all computation is represented on the graph. This enables one to compose random variables with complex deterministic structure such as deep neural networks, a diverse set of math operations, and third party libraries that build on the same framework. The design also enables compositions of random variables to capture complex stochastic structure.

As an illustration, we use a Beta-Bernoulli model, $p(\mathbf{x}, \theta) = \text{Beta}(\theta \mid 1, 1) \prod_{n=1}^{50} \text{Bernoulli}(x_n \mid \theta)$, where $\theta$ is a latent probability shared across the 50 data points $\mathbf{x} \in \{0, 1\}^{50}$. The random variable x is 50-dimensional, parameterized by the random tensor $\theta^*$. Fetching the object x runs the graph: it simulates from the generative process and outputs a binary vector of 50 elements.

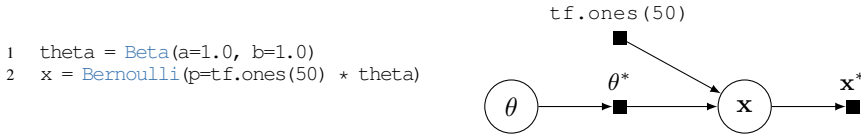

**Figure 1:** Beta-Bernoulli program (**left**) alongside its computational graph (**right**). Fetching $\mathbf{x}$ from the graph generates a binary vector of 50 elements.

All computation is registered symbolically on random variables and not over their execution. Symbolic representations do not require reifying the full model, which leads to unreasonable memory

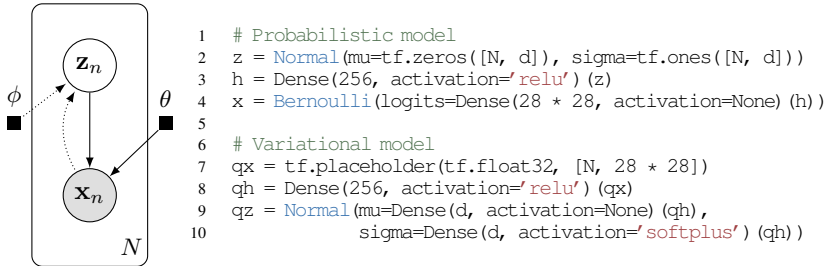

```
1  # Probabilistic model
2  z = Normal(mu=tf.zeros([N, d]), sigma=tf.ones([N, d]))
3  h = Dense(256, activation='relu')(z)
4  x = Bernoulli(logits=Dense(28 * 28, activation=None)(h))
5
6  # Variational model
7  qx = tf.placeholder(tf.float32, [N, 28 * 28])
8  qh = Dense(256, activation='relu')(qx)
9  qz = Normal(mu=Dense(d, activation=None)(qh),
10           sigma=Dense(d, activation='softplus')(qh))
```

**Figure 2:** Variational auto-encoder for a data set of $28 \times 28$ pixel images: **(left)** graphical model, with dotted lines for the inference model; **(right)** probabilistic program, with 2-layer neural networks.

consumption for large models (Tristan et al., 2014). Moreover, it enables us to simplify both deterministic and stochastic operations in the graph, before executing any code (Ścibior et al., 2015; Zinkov & Shan, 2016).

With computational graphs, it is also natural to build mutable states within the probabilistic program. As a typical use of computational graphs, such states can define model parameters; in TensorFlow, this is given by a `tf.Variable`. Another use case is for building discriminative models $p(\mathbf{y} \mid \mathbf{x})$, where $\mathbf{x}$ are features that are input as training or test data. The program can be written independent of the data, using a mutable state (`tf.placeholder`) for $\mathbf{x}$ in its graph. During training and testing, we feed the placeholder the appropriate values.

In Appendix A, we provide examples of a Bayesian neural network for classification (A.1), latent Dirichlet allocation (A.2), and Gaussian matrix factorization (A.3). We present others below.

### 3.1 EXAMPLE: VARIATIONAL AUTO-ENCODER

Figure 2 implements a variational auto-encoder (VAE) (Kingma & Welling, 2014; Rezende et al., 2014) in Edward. It comprises a probabilistic model over data and a variational model designed to approximate the former's posterior. Here we use random variables to construct both the probabilistic model and the variational model; they are fit during inference (more details in Section 4).

There are $N$ data points $x_n \in \{0, 1\}^{28 \cdot 28}$ each with $d$ latent variables, $z_n \in \mathbb{R}^d$. The program uses Keras (Chollet, 2015) to define neural networks. The probabilistic model is parameterized by a 2-layer neural network, with 256 hidden units (and ReLU activation), and generates $28 \times 28$ pixel images. The variational model is parameterized by a 2-layer inference network, with 256 hidden units and outputs parameters of a normal posterior approximation.

The probabilistic program is concise. Core elements of the VAE—such as its distributional assumptions and neural net architectures—are all extensible. With model compositionality, we can embed it into more complicated models (Gregor et al., 2015; Rezende et al., 2016) and for other learning tasks (Kingma et al., 2014). With inference compositionality (which we discuss in Section 4), we can embed it into more complicated algorithms, such as with expressive variational approximations (Rezende & Mohamed, 2015; Tran et al., 2016b; Kingma et al., 2016) and alternative objectives (Ranganath et al., 2016a; Li & Turner, 2016; Dieng et al., 2016).

### 3.2 EXAMPLE: BAYESIAN RECURRENT NEURAL NETWORK WITH VARIABLE LENGTH

Random variables can also be composed with control flow operations. As an example, Figure 3 implements a Bayesian recurrent neural network (RNN) with variable length. The data is a sequence of inputs $\{\mathbf{x}_1, \ldots, \mathbf{x}_T\}$ and outputs $\{y_1, \ldots, y_T\}$ of length $T$ with $\mathbf{x}_t \in \mathbb{R}^D$ and $y_t \in \mathbb{R}$ per time step. For $t = 1, \ldots, T$, a RNN applies the update

$$\mathbf{h}_t = \tanh(\mathbf{W}_h \mathbf{h}_{t-1} + \mathbf{W}_x \mathbf{x}_t + \mathbf{b}_h),$$

where the previous hidden state is $\mathbf{h}_{t-1} \in \mathbb{R}^H$. We feed each hidden state into the output's likelihood, $y_t \sim \text{Normal}(\mathbf{W}_y \mathbf{h}_t + \mathbf{b}_y, 1)$, and we place a standard normal prior over all parameters $\{\mathbf{W}_h \in \mathbb{R}^{H \times H}, \mathbf{W}_x \in \mathbb{R}^{D \times H}, \mathbf{W}_y \in \mathbb{R}^{H \times 1}, \mathbf{b}_h \in \mathbb{R}^H, \mathbf{b}_y \in \mathbb{R}\}$. Our implementation is dynamic: it differs from a RNN with fixed length, which pads and unrolls the computation.

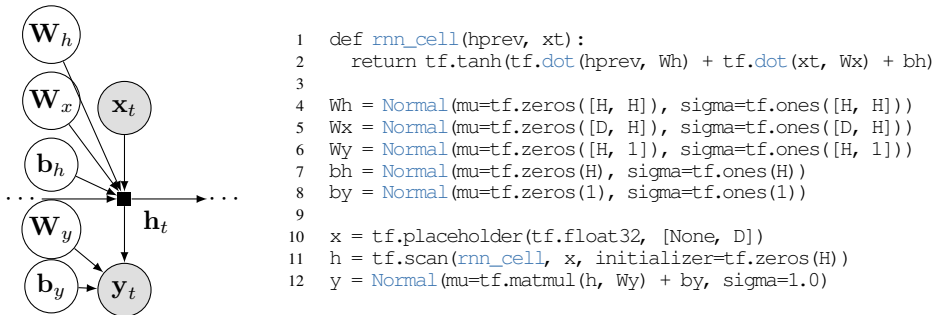

```
1   def rnn_cell(hprev, xt):
2       return tf.tanh(tf.dot(hprev, Wh) + tf.dot(xt, Wx) + bh)
3
4   Wh = Normal(mu=tf.zeros([H, H]), sigma=tf.ones([H, H]))
5   Wx = Normal(mu=tf.zeros([D, H]), sigma=tf.ones([D, H]))
6   Wy = Normal(mu=tf.zeros([H, 1]), sigma=tf.ones([H, 1]))
7   bh = Normal(mu=tf.zeros(H), sigma=tf.ones(H))
8   by = Normal(mu=tf.zeros(1), sigma=tf.ones(1))
9
10  x = tf.placeholder(tf.float32, [None, D])
11  h = tf.scan(rnn_cell, x, initializer=tf.zeros(H))
12  y = Normal(mu=tf.matmul(h, Wy) + by, sigma=1.0)
```

**Figure 3:** Bayesian RNN: **(left)** graphical model; **(right)** probabilistic program. The program has an unspecified number of time steps; it uses a symbolic for loop (`tf.scan`).

## 3.3 Stochastic Control Flow and Model Parallelism

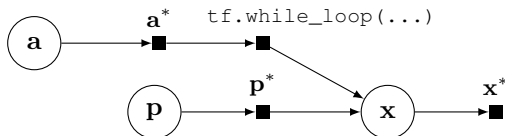

**Figure 4:** Computational graph for a probabilistic program with stochastic control flow.

Random variables can also be placed in the control flow itself, enabling probabilistic programs with stochastic control flow. Stochastic control flow defines dynamic conditional dependencies, known in the literature as contingent or existential dependencies (Mansinghka et al., 2014; Wu et al., 2016). See Figure 4, where $\mathbf{x}$ may or may not depend on $\mathbf{a}$ for a given execution. In Appendix A.4, we use stochastic control flow to implement a Dirichlet process mixture model. Tensors with stochastic shape are also possible: for example, `tf.zeros(Poisson(lam=5.0))` defines a vector of zeros with length given by a Poisson draw with rate $5.0$.

Stochastic control flow produces difficulties for algorithms that use the graph structure because the relationship of conditional dependencies changes across execution traces. The computational graph, however, provides an elegant way of teasing out static conditional dependence structure ($\mathbf{p}$) from dynamic dependence structure ($\mathbf{a}$). We can perform model parallelism (parallel computation across components of the model) over the static structure with GPUs and batch training. We can use more generic computations to handle the dynamic structure.

## 4 Compositional Representations for Inference

We described random variables as a representation for building rich probabilistic programs over computational graphs. We now describe a compositional representation for inference. We desire two criteria: (a) support for many classes of inference, where the form of the inferred posterior depends on the algorithm; and (b) invariance of inference under the computational graph, that is, the posterior can be further composed as part of another model.

To explain our approach, we will use a simple hierarchical model as a running example. Figure 5 displays a joint distribution $p(\mathbf{x}, \mathbf{z}, \beta)$ of data $\mathbf{x}$, local variables $\mathbf{z}$, and global variables $\beta$. The ideas here extend to more expressive programs.

### 4.1 Inference as Stochastic Graph Optimization

The goal of inference is to calculate the posterior distribution $p(\mathbf{z}, \beta \mid \mathbf{x}_{\text{train}}; \boldsymbol{\theta})$ given data $\mathbf{x}_{\text{train}}$, where $\boldsymbol{\theta}$ are any model parameters that we will compute point estimates for.[2] We formalize this as

---

[2]For example, we could replace `x`'s `sigma` argument with `tf.exp(tf.Variable(0.0))*tf.ones([N, D])`. This defines a model parameter initialized at 0 and positive-constrained.

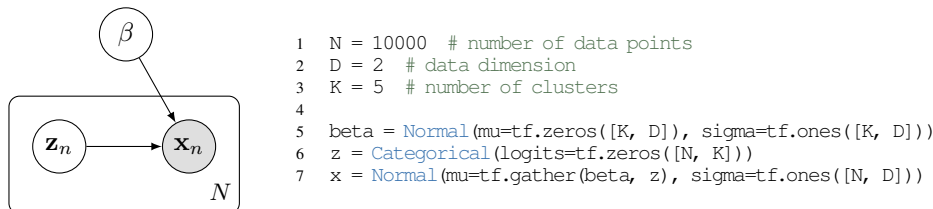

```
1   N = 10000  # number of data points
2   D = 2  # data dimension
3   K = 5  # number of clusters
4
5   beta = Normal(mu=tf.zeros([K, D]), sigma=tf.ones([K, D]))
6   z = Categorical(logits=tf.zeros([N, K]))
7   x = Normal(mu=tf.gather(beta, z), sigma=tf.ones([N, D]))
```

**Figure 5:** Hierarchical model: **(left)** graphical model; **(right)** probabilistic program. It is a mixture of Gaussians over $D$-dimensional data $\{x_n\} \in \mathbb{R}^{N \times D}$. There are $K$ latent cluster means $\beta \in \mathbb{R}^{K \times D}$.

the following optimization problem:

$$\min_{\boldsymbol{\lambda}, \boldsymbol{\theta}} \mathcal{L}(p(\mathbf{z}, \beta \mid \mathbf{x}_{\text{train}}; \boldsymbol{\theta}),\ q(\mathbf{z}, \beta; \boldsymbol{\lambda})), \tag{1}$$

where $q(\mathbf{z}, \beta; \boldsymbol{\lambda})$ is an approximation to the posterior $p(\mathbf{z}, \beta \mid \mathbf{x}_{\text{train}}; \boldsymbol{\theta})$, and $\mathcal{L}$ is a loss function with respect to $p$ and $q$.

The choice of approximation $q$, loss $\mathcal{L}$, and rules to update parameters $\{\boldsymbol{\theta}, \boldsymbol{\lambda}\}$ are specified by an inference algorithm. (Note $q$ can be nonparametric, such as a point or a collection of samples.)

In Edward, we write this problem as follows:

```
1   inference = ed.Inference({beta: qbeta, z: qz}, data={x: x_train})
```

`Inference` is an abstract class which takes two inputs. The first is a collection of latent random variables `beta` and `z`, associated to their "posterior variables" `qbeta` and `qz` respectively. The second is a collection of observed random variables `x`, which is associated to their realizations `x_train`.

The idea is that `Inference` defines and solves the optimization in Equation 1. It adjusts parameters of the distribution of `qbeta` and `qz` (and any model parameters) to be close to the posterior.

Class methods are available to finely control the inference. Calling `inference.initialize()` builds a computational graph to update $\{\boldsymbol{\theta}, \boldsymbol{\lambda}\}$. Calling `inference.update()` runs this computation once to update $\{\boldsymbol{\theta}, \boldsymbol{\lambda}\}$; we call the method in a loop until convergence. Importantly, no efficiency is lost in Edward's language: the computational graph is the same as if it were handwritten for a specific model. This means the runtime is the same; also see our experiments in Section 5.2.

A key concept in Edward is that there is no distinct "model" or "inference" block. A model is simply a collection of random variables, and inference is a way of modifying parameters in that collection subject to another. This reductionism offers significant flexibility. For example, we can infer only parts of a model (e.g., layer-wise training (Hinton et al., 2006)), infer parts used in multiple models (e.g., multi-task learning), or plug in a posterior into a new model (e.g., Bayesian updating).

## 4.2 CLASSES OF INFERENCE

The design of `Inference` is very general. We describe subclasses to represent many algorithms below: variational inference, Monte Carlo, and generative adversarial networks.

Variational inference posits a family of approximating distributions and finds the closest member in the family to the posterior (Jordan et al., 1999). In Edward, we build the variational family in the graph; see Figure 6 (left). For our running example, the family has mutable variables as parameters $\boldsymbol{\lambda} = \{\pi, \mu, \sigma\}$, where $q(\beta; \mu, \sigma) = \text{Normal}(\beta; \mu, \sigma)$ and $q(\mathbf{z}; \pi) = \text{Categorical}(\mathbf{z}; \pi)$.

Specific variational algorithms inherit from the `VariationalInference` class. Each defines its own methods, such as a loss function and gradient. For example, we represent maximum a posteriori (MAP) estimation with an approximating family (`qbeta` and `qz`) of `PointMass` random variables, i.e., with all probability mass concentrated at a point. `MAP` inherits from `VariationalInference` and defines the negative log joint density as the loss function; it uses existing optimizers inside TensorFlow. In Section 5.1, we experiment with multiple gradient estimators for black box variational inference (Ranganath et al., 2014). Each estimator implements the same loss (an objective proportional to the divergence $\text{KL}(q \,\|\, p)$) and a different update rule (stochastic gradient).

```
1  qbeta = Normal(                              1  T = 10000  # number of samples
2    mu=tf.Variable(tf.zeros([K, D])),          2  qbeta = Empirical(
3    sigma=tf.exp(tf.Variable(tf.zeros([K, D])))) 3    params=tf.Variable(tf.zeros([T, K, D])))
4  qz = Categorical(                            4  qz = Empirical(
5    logits=tf.Variable(tf.zeros([N, K])))      5    params=tf.Variable(tf.zeros([T, N])))
6                                               6
7  inference = ed.VariationalInference(         7  inference = ed.MonteCarlo(
8    {beta: qbeta, z: qz}, data={x: x_train})   8    {beta: qbeta, z: qz}, data={x: x_train})
```

**Figure 6: (left)** Variational inference. **(right)** Monte Carlo.

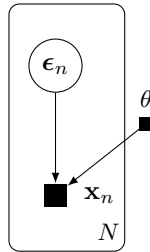

```
1   def generative_network(eps):
2       h = Dense(256, activation='relu')(eps)
3       return Dense(28 * 28, activation=None)(h)
4
5   def discriminative_network(x):
6       h = Dense(28 * 28, activation='relu')(x)
7       return Dense(1, activation=None)(h)
8
9   # Probabilistic model
10  eps = Normal(mu=tf.zeros([N, d]), sigma=tf.ones([N, d]))
11  x = generative_network(eps)
12
13  inference = ed.GANInference(data={x: x_train},
14      discriminator=discriminative_network)
```

**Figure 7:** Generative adversarial networks: **(left)** graphical model; **(right)** probabilistic program. The model (generator) uses a parameterized function (discriminator) for training.

Monte Carlo approximates the posterior using samples (Robert & Casella, 1999). Monte Carlo is an inference where the approximating family is an empirical distribution, $q(\beta; \{\beta^{(t)}\}) = \frac{1}{T} \sum_{t=1}^{T} \delta(\beta, \beta^{(t)})$ and $q(\mathbf{z}; \{\mathbf{z}^{(t)}\}) = \frac{1}{T} \sum_{t=1}^{T} \delta(\mathbf{z}, \mathbf{z}^{(t)})$. The parameters are $\boldsymbol{\lambda} = \{\beta^{(t)}, \mathbf{z}^{(t)}\}$. See Figure 6 (right). Monte Carlo algorithms proceed by updating one sample $\beta^{(t)}, \mathbf{z}^{(t)}$ at a time in the empirical approximation. Specific MC samplers determine the update rules: they can use gradients such as in Hamiltonian Monte Carlo (Neal, 2011) and graph structure such as in sequential Monte Carlo (Doucet et al., 2001).

Edward also supports non-Bayesian methods such as generative adversarial networks (GANs) (Goodfellow et al., 2014). See Figure 7. The model posits random noise `eps` over $N$ data points, each with $d$ dimensions; this random noise feeds into a `generative_network` function, a neural network that outputs real-valued data `x`. In addition, there is a `discriminative_network` which takes data as input and outputs the probability that the data is real (in logit parameterization). We build `GANInference`; running it optimizes parameters inside the two neural network functions. This approach extends to many advances in GANs (e.g., Denton et al. (2015); Li et al. (2015)).

Finally, one can design algorithms that would otherwise require tedious algebraic manipulation. With symbolic algebra on nodes of the computational graph, we can uncover conjugacy relationships between random variables. Users can then integrate out variables to automatically derive classical Gibbs (Gelfand & Smith, 1990), mean-field updates (Bishop, 2006), and exact inference. These algorithms are being currently developed in Edward.

## 4.3 COMPOSING INFERENCES

Core to Edward's design is that inference can be written as a collection of separate inference programs. Below we demonstrate variational EM, with an (approximate) E-step over local variables and an M-step over global variables. We instantiate two algorithms, each of which conditions on inferences from the other, and we alternate with one update of each (Neal & Hinton, 1993),

```
1  qbeta = PointMass(params=tf.Variable(tf.zeros([K, D])))
2  qz = Categorical(logits=tf.Variable(tf.zeros([N, K])))
3
4  inference_e = ed.VariationalInference({z: qz}, data={x: x_train, beta: qbeta})
5  inference_m = ed.MAP({beta: qbeta}, data={x: x_train, z: qz})
6  ...
```

```
7  for _ in range(10000):
8    inference_e.update()
9    inference_m.update()
```

This extends to many other cases such as exact EM for exponential families, contrastive divergence (Hinton, 2002), pseudo-marginal methods (Andrieu & Roberts, 2009), and Gibbs sampling within variational inference (Wang & Blei, 2012; Hoffman & Blei, 2015). We can also write message passing algorithms, which solve a collection of local inference problems (Koller & Friedman, 2009). For example, classical message passing uses exact local inference and expectation propagation locally minimizes the Kullback-Leibler divergence, $\mathrm{KL}(p \,\|\, q)$ (Minka, 2001).

## 4.4 Data Subsampling

Stochastic optimization (Bottou, 2010) scales inference to massive data and is key to algorithms such as stochastic gradient Langevin dynamics (Welling & Teh, 2011) and stochastic variational inference (Hoffman et al., 2013). The idea is to cheaply estimate the model's log joint density in an unbiased way. At each step, one subsamples a data set $\{x_m\}$ of size $M$ and then scales densities with respect to local variables,

$$\log p(\mathbf{x}, \mathbf{z}, \beta) = \log p(\beta) + \sum_{n=1}^{N} \Big[ \log p(x_n \,|\, z_n, \beta) + \log p(z_n \,|\, \beta) \Big]$$

$$\approx \log p(\beta) + \frac{N}{M} \sum_{m=1}^{M} \Big[ \log p(x_m \,|\, z_m, \beta) + \log p(z_m \,|\, \beta) \Big].$$

To support stochastic optimization, we represent only a subgraph of the full model. This prevents reifying the full model, which can lead to unreasonable memory consumption (Tristan et al., 2014). During initialization, we pass in a dictionary to properly scale the arguments. See Figure 8.

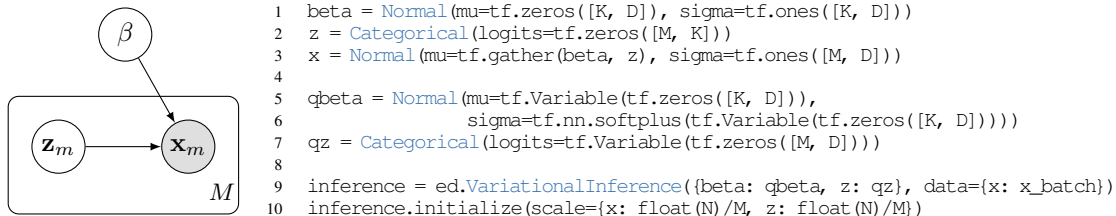

```
1   beta = Normal(mu=tf.zeros([K, D]), sigma=tf.ones([K, D]))
2   z = Categorical(logits=tf.zeros([M, K]))
3   x = Normal(mu=tf.gather(beta, z), sigma=tf.ones([M, D]))
4
5   qbeta = Normal(mu=tf.Variable(tf.zeros([K, D])),
6              sigma=tf.nn.softplus(tf.Variable(tf.zeros([K, D]))))
7   qz = Categorical(logits=tf.Variable(tf.zeros([M, D])))
8
9   inference = ed.VariationalInference({beta: qbeta, z: qz}, data={x: x_batch})
10  inference.initialize(scale={x: float(N)/M, z: float(N)/M})
```

**Figure 8:** Data subsampling with a hierarchical model. We define a subgraph of the full model, forming a plate of size $M$ rather than $N$. We then scale all local random variables by $N/M$.

Conceptually, the scale argument represents scaling for each random variable's plate, as if we had seen that random variable $N/M$ as many times. As an example, Appendix B shows how to implement stochastic variational inference in Edward. The approach extends naturally to streaming data (Doucet et al., 2000; Broderick et al., 2013; McInerney et al., 2015), dynamic batch sizes, and data structures in which working on a subgraph does not immediately apply (Binder et al., 1997; Johnson & Willsky, 2014; Foti et al., 2014).

## 5 Experiments

In this section, we illustrate two main benefits of Edward: flexibility and efficiency. For the former, we show how it is easy to compare different inference algorithms on the same model. For the latter, we show how it is easy to get significant speedups by exploiting computational graphs.

## 5.1 Recent Methods in Variational Inference

We demonstrate Edward's flexibility for experimenting with complex inference algorithms. We consider the VAE setup from Figure 2 and the binarized MNIST data set (Salakhutdinov & Murray,

| Inference method | Negative log-likelihood |
|---|---|
| VAE (Kingma & Welling, 2014) | $\leq 88.2$ |
| VAE without analytic KL | $\leq 89.4$ |
| VAE with analytic entropy | $\leq 88.1$ |
| VAE with score function gradient | $\leq 87.9$ |
| Normalizing flows (Rezende & Mohamed, 2015) | $\leq 85.8$ |
| Hierarchical variational model (Ranganath et al., 2016b) | $\leq 85.4$ |
| Importance-weighted auto-encoders ($K = 50$) (Burda et al., 2016) | $\leq 86.3$ |
| HVM with IWAE objective ($K = 5$) | $\leq 85.2$ |
| Rényi divergence ($\alpha = -1$) (Li & Turner, 2016) | $\leq 140.5$ |

**Table 1:** Inference methods for a probabilistic decoder on binarized MNIST. The Edward PPL is a convenient research platform, making it easy to both develop and experiment with many algorithms.

2008). We use $d = 50$ latent variables per data point and optimize using ADAM. We study different components of the VAE setup using different methods; Appendix C.1 is a complete script. After training we evaluate held-out log likelihoods, which are lower bounds on the true value.

Table 1 shows the results. The first method uses the VAE from Figure 2. The next three methods use the same VAE but apply different gradient estimators: reparameterization gradient without an analytic KL; reparameterization gradient with an analytic entropy; and the score function gradient (Paisley et al., 2012; Ranganath et al., 2014). This typically leads to the same optima but at different convergence rates. The score function gradient was slowest. Gradients with an analytic entropy produced difficulties around convergence: we switched to stochastic estimates of the entropy as it approached an optima. We also use hierarchical variational models (HVMs) (Ranganath et al., 2016b) with a normalizing flow prior; it produced similar results as a normalizing flow on the latent variable space (Rezende & Mohamed, 2015), and better than importance-weighted auto-encoders (IWAEs) (Burda et al., 2016).

We also study novel combinations, such as HVMs with the IWAE objective, GAN-based optimization on the decoder (with pixel intensity-valued data), and Rényi divergence on the decoder. GAN-based optimization does not enable calculation of the log-likelihood; Rényi divergence does not directly optimize for log-likelihood so it does not perform well. The key point is that Edward is a convenient research platform: they are all easy modifications of a given script.

## 5.2 GPU-ACCELERATED HAMILTONIAN MONTE CARLO

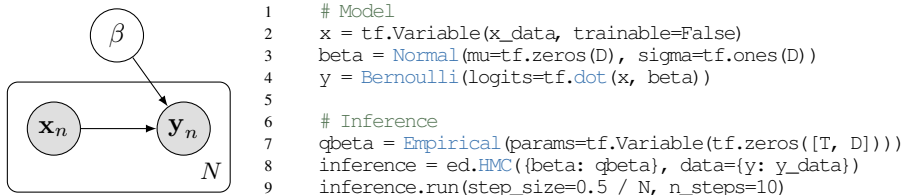

```
1   # Model
2   x = tf.Variable(x_data, trainable=False)
3   beta = Normal(mu=tf.zeros(D), sigma=tf.ones(D))
4   y = Bernoulli(logits=tf.dot(x, beta))
5
6   # Inference
7   qbeta = Empirical(params=tf.Variable(tf.zeros([T, D])))
8   inference = ed.HMC({beta: qbeta}, data={y: y_data})
9   inference.run(step_size=0.5 / N, n_steps=10)
```

**Figure 9:** Edward program for Bayesian logistic regression with Hamiltonian Monte Carlo (HMC).

We benchmark runtimes for a fixed number of Hamiltonian Monte Carlo (HMC; Neal, 2011) iterations on modern hardware: a 12-core Intel i7-5930K CPU at 3.50GHz and an NVIDIA Titan X (Maxwell) GPU. We apply logistic regression on the Covertype dataset ($N = 581012$, $D = 54$; responses were binarized) using Edward, Stan (with PyStan) (Carpenter et al., 2016), and PyMC3 (Salvatier et al., 2015). We ran 100 HMC iterations, with 10 leapfrog updates per iteration, a step size of $0.5/N$, and single precision. Figure 9 illustrates the program in Edward.

Table 2 displays the runtimes.[3] Edward (GPU) features a dramatic 35x speedup over Stan (1 CPU) and 6x speedup over PyMC3 (12 CPU). This showcases the value of building a PPL on top of com-

---

[3]In a previous version of this paper, we reported PyMC3 took 361s. This was caused by a bug preventing PyMC3 from correctly handling single-precision floating point. (PyMC3 with double precision is roughly 14x

| Probabilistic programming system | Runtime (s) |
|---|---|
| Handwritten NumPy (1 CPU) | 534 |
| Stan (1 CPU) (Carpenter et al., 2016) | 171 |
| PyMC3 (12 CPU) (Salvatier et al., 2015) | 30.0 |
| **Edward (12 CPU)** | **8.2** |
| Handwritten TensorFlow (GPU) | 5.0 |
| **Edward (GPU)** | **4.9** |

**Table 2:** HMC benchmark for large-scale logistic regression. Edward (GPU) is significantly faster than other systems. In addition, Edward has no overhead: it is as fast as handwritten TensorFlow.

putational graphs. The speedup stems from fast matrix multiplication when calculating the model's log-likelihood; GPUs can efficiently parallelize this computation. We expect similar speedups for models whose bottleneck is also matrix multiplication, such as deep neural networks.

There are various reasons for the speedup. Stan only used 1 CPU as it leverages multiple cores by running HMC chains in parallel. Stan also used double-precision floating point as it does not allow single-precision. For PyMC3, we note Edward's speedup is not a result of PyMC3's Theano backend compared to Edward's TensorFlow. Rather, PyMC3 does not use Theano for all its computation, so it experiences communication overhead with NumPy. (PyMC3 was actually slower when using the GPU.) We predict that porting Edward's design to Theano would feature similar speedups.

In addition to these speedups, we highlight that Edward has no runtime overhead: it is as fast as handwritten TensorFlow. Following Section 4.1, this is because the computational graphs for inference are in fact the same for Edward and the handwritten code.

### 5.3 PROBABILITY ZOO

In addition to Edward, we also release the *Probability Zoo*, a community repository for pre-trained probability models and their posteriors.[4] It is inspired by the model zoo in Caffe (Jia et al., 2014), which provides many pre-trained discriminative neural networks, and which has been key to making large-scale deep learning more transparent and accessible. It is also inspired by Forest (Stuhlmüller, 2012), which provides examples of probabilistic programs.

## 6 DISCUSSION: CHALLENGES & EXTENSIONS

We described Edward, a Turing-complete PPL with compositional representations for probabilistic models and inference. Edward expands the scope of probabilistic programming to be as flexible and computationally efficient as traditional deep learning. For flexibility, we showed how Edward can use a variety of composable inference methods, capture recent advances in variational inference and generative adversarial networks, and finely control the inference algorithms. For efficiency, we showed how Edward leverages computational graphs to achieve fast, parallelizable computation, scales to massive data, and incurs no runtime overhead over handwritten code.

In present work, we are applying Edward as a research platform for developing new probabilistic models (Rudolph et al., 2016; Tran et al., 2017) and new inference algorithms (Dieng et al., 2016). As with any language design, Edward makes tradeoffs in pursuit of its flexibility and speed for research. For example, an open challenge in Edward is to better facilitate programs with complex control flow and recursion. While possible to represent, it is unknown how to enable their flexible inference strategies. In addition, it is open how to expand Edward's design to dynamic computational graph frameworks—which provide more flexibility in their programming paradigm—but may sacrifice performance. A crucial next step for probabilistic programming is to leverage dynamic computational graphs while maintaining the flexibility and efficiency that Edward offers.

---

slower than Edward (GPU).) This has been fixed after discussion with Thomas Wiecki. The reported numbers also exclude compilation time, which is significant for Stan.

[4]The Probability Zoo is available at http://edwardlib.org/zoo. It includes model parameters and inferred posterior factors, such as local and global variables during training and any inference networks.

ACKNOWLEDGEMENTS

We thank the probabilistic programming community—for sharing our enthusiasm and motivating further work—including developers of Church, Venture, Gamalon, Hakaru, and WebPPL. We also thank Stan developers for providing extensive feedback as we developed the language, as well as Thomas Wiecki for experimental details. We thank the Google BayesFlow team—Joshua Dillon, Ian Langmore, Ryan Sepassi, and Srinivas Vasudevan—as well as Amr Ahmed, Matthew Johnson, Hung Bui, Rajesh Ranganath, Maja Rudolph, and Francisco Ruiz for their helpful feedback. This work is supported by NSF IIS-1247664, ONR N00014-11-1-0651, DARPA FA8750-14-2-0009, DARPA N66001-15-C-4032, Adobe, Google, NSERC PGS-D, and the Sloan Foundation.

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

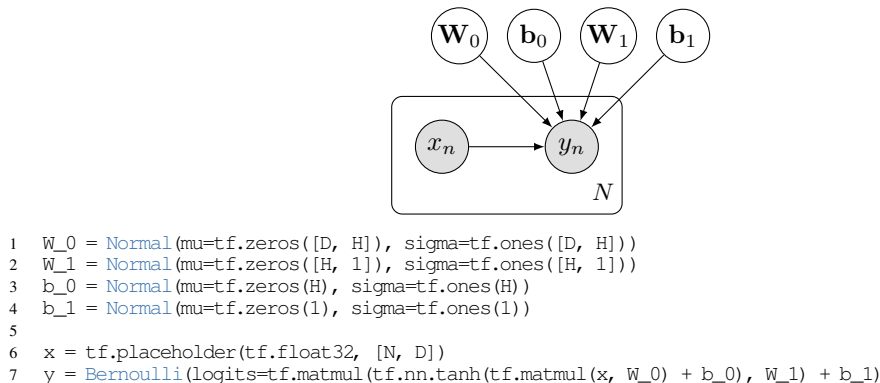

```
1  W_0 = Normal(mu=tf.zeros([D, H]), sigma=tf.ones([D, H]))
2  W_1 = Normal(mu=tf.zeros([H, 1]), sigma=tf.ones([H, 1]))
3  b_0 = Normal(mu=tf.zeros(H), sigma=tf.ones(H))
4  b_1 = Normal(mu=tf.zeros(1), sigma=tf.ones(1))
5
6  x = tf.placeholder(tf.float32, [N, D])
7  y = Bernoulli(logits=tf.matmul(tf.nn.tanh(tf.matmul(x, W_0) + b_0), W_1) + b_1)
```

**Figure 10:** Bayesian neural network for classification.

# A    MODEL EXAMPLES

There are many examples available at http://edwardlib.org, including models, inference methods, and complete scripts. Below we describe several model examples; Appendix B describes an inference example (stochastic variational inference); Appendix C describes complete scripts. All examples in this paper are comprehensive, only leaving out import statements and fixed values. See the companion webpage for this paper (http://edwardlib.org/iclr2017) for examples in a machine-readable format with runnable code.

## A.1    BAYESIAN NEURAL NETWORK FOR CLASSIFICATION

A Bayesian neural network is a neural network with a prior distribution on its weights.

Define the likelihood of an observation $(\mathbf{x}_n, y_n)$ with binary label $y_n \in \{0, 1\}$ as

$$p(y_n \mid \mathbf{W}_0, \mathbf{b}_0, \mathbf{W}_1, \mathbf{b}_1 \; ; \; \mathbf{x}_n) = \text{Bernoulli}(y_n \mid \text{NN}(\mathbf{x}_n \; ; \; \mathbf{W}_0, \mathbf{b}_0, \mathbf{W}_1, \mathbf{b}_1)),$$

where NN is a 2-layer neural network whose weights and biases form the latent variables $\mathbf{W}_0, \mathbf{b}_0, \mathbf{W}_1, \mathbf{b}_1$. Define the prior on the weights and biases to be the standard normal. See Figure 10. There are $N$ data points, $D$ features, and $H$ hidden units.

## A.2    LATENT DIRICHLET ALLOCATION

See Figure 11. Note that the program is written for illustration. We recommend vectorization in practice: instead of storing scalar random variables in lists of lists, one should prefer to represent few random variables, each which have many dimensions.

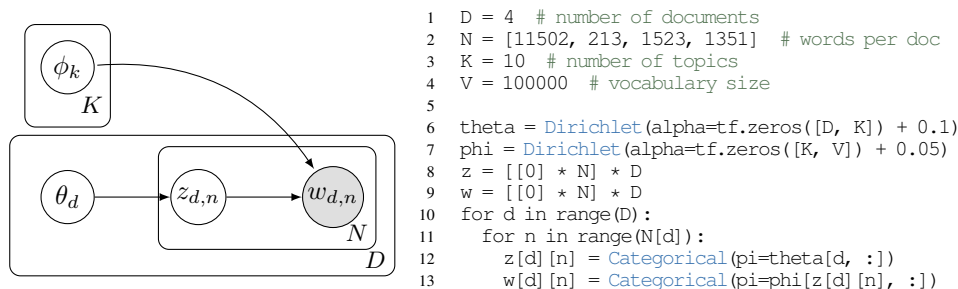

```
1   D = 4   # number of documents
2   N = [11502, 213, 1523, 1351]   # words per doc
3   K = 10   # number of topics
4   V = 100000   # vocabulary size
5
6   theta = Dirichlet(alpha=tf.zeros([D, K]) + 0.1)
7   phi = Dirichlet(alpha=tf.zeros([K, V]) + 0.05)
8   z = [[0] * N] * D
9   w = [[0] * N] * D
10  for d in range(D):
11    for n in range(N[d]):
12      z[d][n] = Categorical(pi=theta[d, :])
13      w[d][n] = Categorical(pi=phi[z[d][n], :])
```

**Figure 11:** Latent Dirichlet allocation (Blei et al., 2003).

## A.3 GAUSSIAN MATRIX FACTORIZATIONN

See Figure 12.

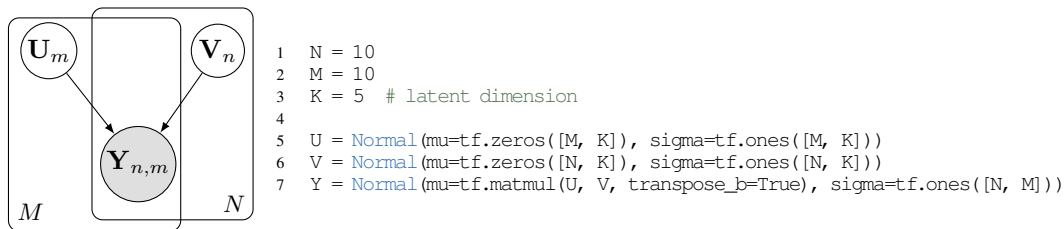

```
1   N = 10
2   M = 10
3   K = 5   # latent dimension
4
5   U = Normal(mu=tf.zeros([M, K]), sigma=tf.ones([M, K]))
6   V = Normal(mu=tf.zeros([N, K]), sigma=tf.ones([N, K]))
7   Y = Normal(mu=tf.matmul(U, V, transpose_b=True), sigma=tf.ones([N, M]))
```

**Figure 12:** Gaussian matrix factorization.

## A.4 DIRICHLET PROCESS MIXTURE MODEL

See Figure 13.

A Dirichlet process mixture model is written as follows:

```
1   mu = DirichletProcess(alpha=0.1, base_cls=Normal, mu=tf.zeros(D), sigma=tf.ones(D), sample_n=N)
2   x = Normal(mu=mu, sigma=tf.ones([N, D]))
```

where `mu` has shape `(N, D)`. The `DirichletProcess` random variable returns `sample_n=N` draws, each with shape given by the base distribution `Normal(mu, sigma)`. The essential component defining the `DirichletProcess` random variable is a stochastic while loop. We define it below. See Edward's code base for a more involved version with a base distribution.

```
1   def dirichlet_process(alpha):
2     def cond(k, beta_k):
3       flip = Bernoulli(p=beta_k)
4       return tf.equal(flip, tf.constant(1))
5
6     def body(k, beta_k):
7       beta_k = beta_k * Beta(a=1.0, b=alpha)
8       return k + 1, beta_k
9
10    k = tf.constant(0)
11    beta_k = Beta(a=1.0, b=alpha)
12    stick_num, stick_beta = tf.while_loop(cond, body, loop_vars=[k, beta_k])
13    return stick_num
```

**Figure 13:** Dirichlet process mixture model.

## B INFERENCE EXAMPLE: STOCHASTIC VARIATIONAL INFERENCE

In the subgraph setting, we do data subsampling while working with a subgraph of the full model. This setting is necessary when the data and model do not fit in memory. It is scalable in that both the algorithm's computational complexity (per iteration) and memory complexity are independent of the data set size.

For the code, we use the running example, a mixture model described in Figure 5.

```
1   N = 10000000  # data set size
2   D = 2   # data dimension
3   K = 5   # number of clusters
```

The model is

$$p(\mathbf{x}, \mathbf{z}, \beta) = p(\beta) \prod_{n=1}^{N} p(z_n \mid \beta) p(x_n \mid z_n, \beta).$$

To avoid memory issues, we work on only a subgraph of the model,

$$p(\mathbf{x}, \mathbf{z}, \beta) = p(\beta) \prod_{m=1}^{M} p(z_m \mid \beta) p(x_m \mid z_m, \beta)$$

```
1  M = 128  # mini-batch size
2
3  beta = Normal(mu=tf.zeros([K, D]), sigma=tf.ones([K, D]))
4  z = Categorical(logits=tf.zeros([M, K]))
5  x = Normal(mu=tf.gather(beta, z), sigma=tf.ones([M, D]))
```

Assume the variational model is

$$q(\mathbf{z}, \beta) = q(\beta; \lambda) \prod_{n=1}^{N} q(z_n \mid \beta; \gamma_n),$$

parameterized by $\{\lambda, \{\gamma_n\}\}$. Again, we work on only a subgraph of the model,

$$q(\mathbf{z}, \beta) = q(\beta; \lambda) \prod_{m=1}^{M} q(z_m \mid \beta; \gamma_m).$$

parameterized by $\{\lambda, \{\gamma_m\}\}$. Importantly, only $M$ parameters are stored in memory for $\{\gamma_m\}$ rather than $N$.

```
1  qbeta = Normal(mu=tf.Variable(tf.zeros([K, D])),
2              sigma=tf.nn.softplus(tf.Variable(tf.zeros[K, D])))
3  qz_variables = tf.Variable(tf.zeros([M, K]))
4  qz = Categorical(logits=qz_variables)
```

We use `KLqp`, a variational method that minimizes the divergence measure $\text{KL}(q \| p)$ (Jordan et al., 1999). We instantiate two algorithms: a global inference over $\beta$ given the subset of $\mathbf{z}$ and a local inference over the subset of $\mathbf{z}$ given $\beta$. We also pass in a TensorFlow placeholder `x_ph` for the data, so we can change the data at each step.

```
1  x_ph = tf.placeholder(tf.float32, [M])
2  inference_global = ed.KLqp({beta: qbeta}, data={x: x_ph, z: qz})
3  inference_local = ed.KLqp({z: qz}, data={x: x_ph, beta: qbeta})
```

We initialize the algorithms with the `scale` argument, so that computation on `z` and `x` will be scaled appropriately. This enables unbiased estimates for stochastic gradients.

```
1  inference_global.initialize(scale={x: float(N) / M, z: float(N) / M})
2  inference_local.initialize(scale={x: float(N) / M, z: float(N) / M})
```

We now run the algorithm, assuming there is a `next_batch` function which provides the next batch of data.

```
1  qz_init = tf.initialize_variables([qz_variables])
2  for _ in range(1000):
3    x_batch = next_batch(size=M)
4    for _ in range(10):  # make local inferences
5      inference_local.update(feed_dict={x_ph: x_batch})
6
7    # update global parameters
8    inference_global.update(feed_dict={x_ph: x_batch})
9    # reinitialize the local factors
10   qz_init.run()
```

After each iteration, we also reinitialize the parameters for $q(\mathbf{z} \mid \beta)$; this is because we do inference on a new set of local variational factors for each batch. This demo readily applies to other inference algorithms such as `SGLD` (stochastic gradient Langevin dynamics): simply replace `qbeta` and `qz` with `Empirical` random variables; then call `ed.SGLD` instead of `ed.KLqp`.

Note that if the data and model fit in memory but you'd still like to perform data subsampling for fast inference, we recommend not defining subgraphs. You can reify the full model, and simply index the local variables with a placeholder. The placeholder is fed at runtime to determine which of the local variables to update at a time. (For more details, see the website's API.)

## C COMPLETE EXAMPLES

### C.1 VARIATIONAL AUTO-ENCODER

See Figure 14.

```
1   import edward as ed
2   import tensorflow as tf
3
4   from edward.models import Bernoulli, Normal
5   from scipy.misc import imsave
6   from tensorflow.contrib import slim
7   from tensorflow.examples.tutorials.mnist import input_data
8
9   M = 100  # batch size during training
10  d = 2   # latent variable dimension
11
12  # Probability model (subgraph)
13  z = Normal(mu=tf.zeros([M, d]), sigma=tf.ones([M, d]))
14  h = Dense(256, activation='relu')(z)
15  x = Bernoulli(logits=Dense(28 * 28, activation=None)(h))
16
17  # Variational model (subgraph)
18  x_ph = tf.placeholder(tf.float32, [M, 28 * 28])
19  qh = Dense(256, activation='relu')(x_ph)
20  qz = Normal(mu=Dense(d, activation=None)(qh),
21              sigma=Dense(d, activation='softplus')(qh))
22
23  # Bind p(x, z) and q(z | x) to the same TensorFlow placeholder for x.
24  mnist = input_data.read_data_sets("data/mnist", one_hot=True)
25  data = {x: x_ph}
26
27  inference = ed.KLqp({z: qz}, data)
28  optimizer = tf.train.RMSPropOptimizer(0.01, epsilon=1.0)
29  inference.initialize(optimizer=optimizer)
30
31  tf.initialize_all_variables().run()
32
33  n_epoch = 100
34  n_iter_per_epoch = 1000
35  for _ in range(n_epoch):
36    for _ in range(n_iter_per_epoch):
37      x_train, _ = mnist.train.next_batch(M)
38      info_dict = inference.update(feed_dict={x_ph: x_train})
39
40    # Generate images.
41    imgs = x.value().eval()
42    for m in range(M):
43      imsave("img/%d.png" % m, imgs[m].reshape(28, 28))
```

**Figure 14:** Complete script for a VAE (Kingma & Welling, 2014) with batch training. It generates MNIST digits after every 1000 updates.

### C.2 PROBABILISTIC MODEL FOR WORD EMBEDDINGS

See Figure 15. This example uses data subsampling (Section 4.4). The priors and conditional likelihoods are defined only for a minibatch of data. Similarly the variational model only models the embeddings used in a given minibatch. TensorFlow variables contain the embedding vectors for the entire vocabulary. TensorFlow placeholders ensure that the correct embedding vectors are used as variational parameters for a given minibatch.

The Bernoulli variables `y_pos` and `y_neg` are fixed to be 1's and 0's respectively. They model whether a word is indeed the target word for a given context window or has been drawn as a negative sample. Without regularization (via priors), the objective we optimize is identical to negative sampling.

```
1   import edward as ed
2   import tensorflow as tf
3
4   from edward.models import Bernoulli, Normal, PointMass
5
6   N = 581238  # number of total words
7   M = 128  # batch size during training
8   K = 100  # number of factors
9   ns = 3  # number of negative samples
10  cs = 4  # context size
11  L = 50000  # vocabulary size
12
13  # Prior over embedding vectors
14  p_rho = Normal(mu=tf.zeros([M, K]),
15                 sigma=tf.sqrt(N) * tf.ones([M, K]))
16  n_rho = Normal(mu=tf.zeros([M, ns, K]),
17                 sigma=tf.sqrt(N) * tf.ones([M, ns, K]))
18
19  # Prior over context vectors
20  ctx_alphas = Normal(mu=tf.zeros([M, cs, K]),
21                      sigma=tf.sqrt(N)*tf.ones([M, cs, K]))
22
23  # Conditional likelihoods
24  ctx_sum = tf.reduce_sum(ctx_alphas, [1])
25  p_eta = tf.expand_dims(tf.reduce_sum(p_rho * ctx_sum, -1),1)
26  n_eta = tf.reduce_sum(n_rho * tf.tile(tf.expand_dims(ctx_sum, 1), [1, ns, 1]), -1)
27  y_pos = Bernoulli(logits=p_eta)
28  y_neg = Bernoulli(logits=n_eta)
29
30  # placeholders for batch training
31  p_idx = tf.placeholder(tf.int32, [M, 1])
32  n_idx = tf.placeholder(tf.int32, [M, ns])
33  ctx_idx = tf.placeholder(tf.int32, [M, cs])
34
35  # Variational parameters (embedding vectors)
36  rho_params = tf.Variable(tf.random_normal([L, K]))
37  alpha_params = tf.Variable(tf.random_normal([L, K]))
38
39  # Variational distribution on embedding vectors
40  q_p_rho = PointMass(params=tf.squeeze(tf.gather(rho_params, p_idx)))
41  q_n_rho = PointMass(params=tf.gather(rho_params, n_idx))
42  q_alpha = PointMass(params=tf.gather(alpha_params, ctx_idx))
43
44  inference = ed.MAP(
45    {p_rho: q_p_rho, n_rho: q_n_rho, ctx_alphas: q_alpha},
46    data={y_pos: tf.ones((M, 1)), y_neg: tf.zeros((M, ns))})
47
48  inference.initialize()
49  tf.initialize_all_variables().run()
50
51  for _ in range(inference.n_iter):
52    targets, windows, negatives = next_batch(M)  # a function to generate data
53    info_dict = inference.update(feed_dict={p_idx: targets, ctx_idx: windows, n_idx: negatives})
54    inference.print_progress(info_dict)
```

**Figure 15:** Exponential family embedding for binary data (Rudolph et al., 2016). Here, MAP is used to maximize the total sum of conditional log-likelihoods and log-priors.

