# Peer review of "Deep Probabilistic Programming"

_ICLR 2017 — accepted_

[Official Review · AnonReviewer3 · rating 7 · confidence 4 · 14 Dec 2016]
**A significant development to include the flexibility of inference to PPL**
soundness 5 · originality 4 · clarity 3 · impact 4

Thank you for an interesting read.

I found this paper very interesting. Since I don't think (deterministic) approximate inference is separated from the modelling procedure (cf. exact inference), it is important to allow the users to select the inference method to suit their needs and constraints. I'm not an expert of PPL, but to my knowledge this is the first package that I've seen which put more focus on compositional inference. Leveraging tensorflow is also a plus, which allows flexible computation graph design as well as parallel computation using GPUs.

The only question I have is about the design of flexible objective functions to learn hyper-parameters (or in the paper those variables associated with delta q distributions). It seems hyper-parameter learning is also specified as inference, which makes sense if using MAP. However the authors also demonstrated other objective functions such as Renyi divergences, does that mean the user need to define a new class of inference method whenever they want to test an alternative loss function?

[Official Review · AnonReviewer1 · rating 8 · confidence 4 · 15 Dec 2016]
**Very promising probabilistic programming language combining Bayes and deep learning**
originality 2 · clarity 1 · impact 3

The paper introduces Edward, a probabilistic programming language
built over TensorFlow and Python, and supporting a broad range of most
popular contemporary methods in probabilistic machine learning.


Quality:

The Edward library provides an extremely impressive collection of
modern probabilistic inference methods in an easily usable form.
The paper provides a brief review of the most important techniques
especially from a representation learning perspective, combined with
two experiments on implementing various modern variational inference
methods and GPU-accelerated HMC.

The first experiment (variational inference) would be more valuable if
there was a clear link to complete code to reproduce the results
provided. The HMC experiment looks OK, except the characterising Stan
as a hand-optimised implementation seems unfair as the code is clearly
not hand-optimised for this specific model and hardware configuration.
I do not think anyone doubts the quality of your implementation, so
please do not ruin the picture by unsubstantiated sensationalist
claims. Instead of current drama, I would suggest comparing
head-to-head against Stan on single core and separately reporting the
extra speedups you gain from parallelisation and GPU. These numbers
would also help the readers to estimate the performance of the method
for other hardware configurations.


Clarity:

The paper is in general clearly written and easy to read. The numerous
code examples are helpful, but also difficult as it is sometimes
unclear what is missing. It would be very helpful if the authors could
provide and clearly link to a machine-readable companion (a Jupyter
notebook would be great, but even text or HTML would be easier to
copy-paste from than a pdf like the paper) with complete runnable code
for all the examples.


Originality:

The Edward library is clearly a unique collection of probabilistic
inference methods. In terms of the paper, the main threat to novelty
comes from previous publications of the same group. The main paper
refers to Tran et al. (2016a) which covers a lot of similar material,
although from a different perspective. It is unclear if the other
paper has been published or submitted somewhere and if so, where.


Significance:

It seems very likely Edward will have a profound impact on the field
of Bayesian machine learning and deep learning.


Other comments:

In Sec. 2 you draw a clear distinction between specialised languages
(including Stan) and Turing-complete languages such as Edward. This
seems unfair as I believe Stan is also Turing complete. Additionally
no proof is provided to support the Turing-completeness of Edward.

[Official Review · AnonReviewer4 · rating 5 · confidence 4 · 18 Dec 2016]
**Exciting and promising approach which still needs to be demonstrated empirically**

The authors propose a new software package for probabilistic programming, taking advantage of recent successful tools used in the deep learning community. The software looks very promising and has the potential to transform the way we work in the probabilistic modelling community, allowing us to perform rapid-prototyping to iterate through ideas quickly. The composability principles are used insightfully, and the extension of inference to HMC for example, going beyond VI inference (which is simple to implement using existing deep learning tools), makes the software even more compelling. 

However, the most important factor of any PPL is whether it is practical for real-world use cases. This was not demonstrated sufficiently in the submission. There are many example code snippets given in the paper, but most are not evaluated. The Dirichlet process mixture model example (Figure 12) is an important one: do the proposed black-box inference tools really work for this snippet? and will the GAN example (Figure 7) converge when optimised with real data? To convince the community of the practicality of the package it will be necessary to demonstrate these empirically. Currently the only evaluated model is a VAE with various inference techniques, which are not difficult to implement using pure TF.

Presentation:
* Paper presentation could be improved. For example the authors could use more signalling for what is about to be explained. On page 5 qbeta and qz are used without explanation - the authors could mention that an example will be given thereafter.
* I would also suggest to the authors to explain in the preface how the layers are implemented, and how the KL is handled in VI for example.
It will be useful to discuss what values are optimised and what values change as inference is performed (even before section 4.4). This was not clear for the majority of the paper. 

Experiments:
* Why is the run time not reported in table 1?
* What are the "difficulties around convergence" encountered with the analytical entropies? inference issues become more difficult to diagnose as inference is automated. Are there tools to diagnose these with the provided toolbox? 
* Did HMC give sensible results in the experiment at the bottom of page 8? only run time is reported. 
* How difficult is it to get the inference to work (eg HMC) when we don't have full control over the computational graph structure and sampler?
* It would be extremely insightful to give a table comparing the performance (run time, predictive log likelihood, etc) of the various inference tools on more models.
* What benchmarks do you intend to use in the Model Zoo? the difficulty with probabilistic modelling is that there are no set benchmarks over which we can evaluate and compare many models. Model zoo is sensible for the Caffe ecosystem because there exist few benchmarks a large portion of the community was working on (ImageNet for example). What datasets would you use to compare the DPMM on for example?

Minor comments:
* Table 1: I would suggest to compare to Li & Turner with alpha=0.5 (the equivalent of Hellinger distance) as they concluded this value performs best. I'm not sure why alpha=-1 was chosen here. 
* How do you handle discrete distributions (eg Figure 5)?
* x_real is not defined in Figure 7.
* I would suggest highlighting M in Figure 8.
* Comma instead of period after "rized), In" on page 8.

In conclusion I would say that the software developments presented here are quite exciting, and I'm glad the authors are pushing towards practical and accessible "inference for all". In its current form though I am forced to give the submission itself a score of 5.

[Author Response · Dustin Tran · 13 Jan 2017]
**Thank you to the reviewers!**

Thank you to all the reviewers for their insightful feedback. As the reviewers agree, we think Edward "has the potential to transform the way we work in the probabilistic modelling community, allowing us to perform rapid-prototyping to iterate through ideas quickly" (Reviewer 4). We make general comments here; specific comments are made on the reviewer's thread. We have also uploaded a revised submission following the feedback.

A major contribution in the paper is in real-world application: "the Edward library provides an extremely impressive collection of modern probabilistic inference methods in an easily usable form" (Reviewer 1). This enables researchers to build on top of current methods and also easily compare to them as baselines.

A second contribution in the paper is in methodology. We proposed new compositional representations, "putt[ing] focus on compositional inference" (Reviewer 3). This "allow[s] the users to select the inference method to suit their needs and constraints" (Reviewer 3), such as for designing rich variational models and generative adversarial networks. We also show how to integrate such a language into computational graph frameworks; this provides significant speedups with distributing training, parallelism, vectorisation, and GPU support.

Finally, we note that Edward is growing with active contributions from the community (

[Final Decision · Program Chairs · 06 Feb 2017]
**ICLR committee final decision**

There was general agreement from the reviewers that this looks like an important development in the area of probabilistic programming. Some reviewers felt the impact of the work could be very significant. The quality of the work and the paper were perceived as being quite high. The main weakness highlighted by the most negative reviewer (who felt the work was marginally below threshold) is the level of empirical evaluation given within the submitted manuscript. The authors did submit a revision and they outline the reviewerÕs points that they have addressed. It appears that if accepted this manuscript would constitute the first peer-reviewed paper on the subject of this new software package (Edward). Based on both the numeric scores, the quality and potential significance of this work I recommend acceptance.